# The Molecular Evolution of Circadian Clock Genes in Spotted Gar (*Lepisosteus oculatus*)

**DOI:** 10.3390/genes10080622

**Published:** 2019-08-17

**Authors:** Yi Sun, Chao Liu, Moli Huang, Jian Huang, Changhong Liu, Jiguang Zhang, John H. Postlethwait, Han Wang

**Affiliations:** 1School of Biology & Basic Medical Sciences, Medical College of Soochow University, Suzhou 215123, China; 2Center for Circadian Clocks, Soochow University, Suzhou 215123, China; 3Institute of Neuroscience, University of Oregon, Eugene, OR 97403, USA

**Keywords:** circadian clocks, spotted gar, genome duplication, conserved synteny, functional divergence

## Abstract

Circadian rhythms are biological rhythms with a period of approximately 24 h. While canonical circadian clock genes and their regulatory mechanisms appear highly conserved, the evolution of clock gene families is still unclear due to several rounds of whole genome duplication in vertebrates. The spotted gar (*Lepisosteus oculatus*), as a non-teleost ray-finned fish, represents a fish lineage that diverged before the teleost genome duplication (TGD), providing an outgroup for exploring the evolutionary mechanisms of circadian clocks after whole-genome duplication. In this study, we interrogated the spotted gar draft genome sequences and found that spotted gar contains 26 circadian clock genes from 11 families. Phylogenetic analysis showed that 9 of these 11 spotted gar circadian clock gene families have the same number of genes as humans, while the members of the *nfil3* and *cry* families are different between spotted gar and humans. Using phylogenetic and syntenic analyses, we found that *nfil3-1* is conserved in vertebrates, while *nfil3-2* and *nfil3-3* are maintained in spotted gar, teleost fish, amphibians, and reptiles, but not in mammals. Following the two-round vertebrate genome duplication (VGD), spotted gar retained *cry1a*, *cry1b*, and *cry2*, and *cry3* is retained in spotted gar, teleost fish, turtles, and birds, but not in mammals. We hypothesize that duplication of core clock genes, such as (*nfil3* and *cry*), likely facilitated diversification of circadian regulatory mechanisms in teleost fish. We also found that the transcription factor binding element (Ahr::Arnt) is retained only in one of the *per1* or *per2* duplicated paralogs derived from the TGD in the teleost fish, implicating possible subfuctionalization cases. Together, these findings help decipher the repertoires of the spotted gar’s circadian system and shed light on how the vertebrate circadian clock systems have evolved.

## 1. Introduction

Circadian clocks regulate various cellular and physiological activities and processes in organisms ranging from cyanobacteria to mammals, allowing for them to adapt to the day–night cycle on Earth [1]. A circadian oscillator exhibits persistent rhythmical activity with a near 24 h periodicity under constant conditions [2]. The universal mechanisms of circadian clocks are transcriptional/translational feedback loops [3]. In mammals, daily biological rhythms are generated at the molecular level through auto-regulatory positive and negative feedback loops of the core clock genes (e.g., *Clock*, *bmal1*, *Period*, *Cryptochrome*) [4,5]. CLOCK and BMAL1 form a heterodimer (CLOCK: BMAL1) and activate transcription of *Period* and *Cryptochrome* genes by binding to E-box motifs (CACGTG) in their promoter regions. PERIOD and CRYPTOCHROME then form another heterodimer to repress the transcriptional activities of CLOCK: BMAL1 heterodimers [6,7]. The circadian oscillators exhibit remarkable conservation of function across wide evolutionary time spans [1,8,9]. 

Circadian systems have undergone a genetic revolution. Like many other multigene families, the circadian genes that are found in single copies in invertebrates are duplicated in vertebrates [10]. Gene duplication is one of the most important mechanisms in the evolution of gene diversity, presumably because obtaining new functions by modifying preexisting genetic systems is easier than by generating them de novo [11,12]. Investigation of the molecular architecture of traits in vertebrates should take into consideration the past duplication events in this lineage. In humans and mice, *CLOCK1* and *NPAS2*/*CLOCK2* are two known paralogs that were generated by the 2R genome duplication event [13]. Zebrafish and fugu are known to have three copies of the *clock* family (*clock1a*, *clock1b* and *clock2*). Tetraodon has *clock1a* and *clock1b* genes, while medaka and stickleback have *clock1* and *clock2* genes. *clock1a/ clock1b* is are a duplicated pair that resulted from the third round of whole genome duplication that occurred within the ray-finned fish lineage prior to the radiation of the teleost fish (approximately 300 Mya) [13].

Teleost fish are attractive for the study of many evolutionary questions related to diverse aspects of biology. Fish show a remarkable level of diversity in their morphology, ecology, behavior, and genomes, as well as multiple other facets of their biology [14]. Whole genome sequencing analyses of several fish species have shed light on the organization and evolution of fish genomes and now allow for investigation of the evolutionary mechanisms underlying biodiversity in fish lineages [15,16,17]. Teleost genomes differ from mammalian genomes, however, by a whole-genome duplication event, the teleost genome duplication (TGD) [18,19,20,21]. While the TGD allows for the dissection of ancestral gene functions via the partitioning of ancestral subfunctions [22,23], it also obfuscates correlations between teleost disease models and their human counter-parts because of the difficulty of ortholog assignment after the lineage-specific loss of duplicated genes and the asymmetric evolution of gene duplicates [24]. Genomic resources from a ray-finned (Actinopterygian) fish that diverged from teleosts before the TGD would facilitate the connectivity of teleost and mammalian genomes. Analysis of a half dozen genes in spotted gar (*L. o.*), a large, air-breathing ray-finned North American fish, suggested that its lineage has diverged from the teleost lineage before the TGD [25,26]. Thus, spotted gar represents a genomic intermediary between teleost medical models and the human genome, providing a critical link between biological models in teleost fish, to which the spotted gar is biologically similar, and humans, to which spotted gar is genomically similar [16,24].

A major question in molecular evolution is how duplicate genes are retained in a genome. Several studies have attempted to account for the evolutionary fates of duplicate genes [10,22,27,28]. Genetic analyses have identified numerous mammalian homologs of circadian clock gene families, including *Clock*, *Bmal*, *Per*, *Cry*, *Csnk1e*, *Nr1d*, *Ror*, and *Tim*. A number of other transcription factors also thought to function in the circadian regulation of gene expression, including *Dbp*, *Nfil3*, and *Dec* [29,30,31]. We have discovered, in a previous study, that the spotted gar genome retained most of its circadian clock genes, which are similar in copy numbers with those of humans [16]. To further examine the evolution of circadian clock genes in teleost fish, we interrogated the spotted gar draft genome sequences and analyzed 11 families of the main clock genes (including *bmal*, *clock*, *period*, *cry*, *dec*, *csnk1e*, *nr1d*, *ror*, *par*, *nfil3* and *tim*) involved in vertebrates’ circadian rhythm pathways. We found that spotted gar contains 26 circadian clock genes from the 11 families. Nine of these 11 families of spotted gar are the same as those of humans. However, the orthologs of *nfil3* and *cry* are different between spotted gar and humans. These analyses strongly support the notion that spotted gar resemble humans to a great extent, without the TGD, but still contain extra copies of genes that mammals lack.

## 2. Materials and Methods

### 2.1. Data Sets and Phylogenetic Analysis

We selected 11 circadian clock gene families including *period*, *clock*, *bmal*, *cry*, *dec*, *csnk1e*, *nr1d*, *ror*, *par*, *nfil3*, and *tim*. These circadian clock genes from representative species were uncovered from Ensembl (Release 95) (http://www.ensembl.org/index.html). The final data set included 365 gene sequences (Appendix A). Based upon comparative genomic analysis, we have changed the names for some genes including *cry* and *nfil3*, except for *tim* (there is only one *tim* ortholog gene in each species of vertebrate), to better reflect their evolutionary history and increase genome connectivity. All of the old names and corresponding new names of these genes are listed in Appendix A. Multiple sequence alignments of protein sequences were performed with CLUSTAL W [32]. To explore the evolutionary pattern of the circadian rhythm genes, the neighbor-joining (NJ) algorithm was used to construct phylogenetic trees of these circadian clock genes with MEGA6 [33]. Each tree was a consensus tree derived from a heuristic search of 1000 bootstrap replicates. The abbreviated species names are as follows: *Ac*, *Anolis carolinensis*; *Dm*, *Drosophila melanogaster*; *Dr*, *Danio rerio*; *Ga, Gasterosteus aculeatus*; *Gg*, *Gallus gallus*; *Hs*, *Homo sapiens*; *Lc*, *Latimeria chalumnae*; *Lo*, *L. o.*; *Mu*, *Mus musculus*; *Oa*, *Ornithorhynchus anatinus*; *Ol*, *Oryzias latipes*; *On*, *Oreochromis niloticus*; *Pm*, *Petromyzon marinus*; *Ps*, *Pelodiscus sinensis*; *Tg*, *Taeniopygia guttata*; *Tn*, *Tetraodon nigroviridis*; *Tr*, *Takifugu rubripes*; and *Xt*, *Xenopus tropicalis*.

### 2.2. Exon Structural Analysis

Exonic boundaries of the coding regions of these genes were determined according to Ensembl. The exon–intron structures were presented using the FancyGene visualization software (http://bio.ieo.eu/fancygene/) [34]. The function domains were predicted with the online tool InterPro (http://www.ebi.ac.uk/interpro/).

### 2.3. Conserved Synteny Analysis

Gene synteny for the circadian clock genes was compared to the syntenic region using the Genomicus database (http://www.genomicus.biologie.ens.fr/genomicus) [35]. Spotted gar was used as a reference species for cross-species synteny, and because of its protein similarity, to highlight the conservation of synteny and the protein sequence in vertebrates. Synteny data and species images were downloaded from Genomicus. Consistent with the label in the database, the thick blue line between two genes is equivalent to a “gap” in the alignment of this extant species, i.e., the two genes are neighbours in this species but not in the reference species. The thin blue line is equivalent to a “break” in the continuity of the alignment, i.e., the two genes are linked in order, but at least one gene separates them in other species. The dotted line means no alignment, and the double-headed arrow under a block of genes means that the order of the genes shown was flipped around (reversed).

### 2.4. Regulatory Region Analysis

The upstream regions at approximately 3000 bp for *per* genes in vertebrates were obtained from Ensembl. The transcription factor binding elements were predicted using the JASPAR (http://jaspar.genereg.net/) dataset (profile score threshold 99%).

## 3. Results

### 3.1. Phylogenetic Analysis of Spotted Gar Circadian Clock Genes

Using zebrafish and human circadian clock genes to interrogate the spotted gar draft genome sequences, we found that spotted gar contains 26 circadian clock genes from 11 families, which are consistent with our previous report (Table 1) [16]. To help understand how the gar circadian genes link to the human genome, we constructed phylogenetic trees of the 11 families with 365 genes in vertebrates, and showed that eight of these 11 families of spotted gar circadian clock genes, including *period, clock, bmal, tim, dec, csnk1e, nr1d*, and *ror*, have the same number of genes as humans without extra duplicate copies (Figure 1). However, most of the circadian clock families except for *tim* were duplicated in teleost fish. The final data set, including gene sequences, is listed in Appendix A.

For the *nfil3* gene family, spotted gar contains *nfil3-1, nfil3-2*, and *nfil3-3*. While *nfil3-1* is the ortholog of the mammalian *Nfil3* gene, *nfil3-2* and *nfil3-3* were duplicated to *nfil3-2a*/*nfil3-2b*, *nfil3-3a* /*nfil3-3b*, respectively, via TGD in teleost fish, including zebrafish. Some teleost fish possess additional *nfil3-1b.1* and *nfil3-1b.2* genes.

For the *cry* gene family, spotted gar contains two *cry1* genes, *cry1a*, and *cry1b*, which are co-orthologs of mammalian *Cry1,* likely derived from tandem duplication in teleost fish. Zebrafish have four *cry1* genes, *cry1aa*, *cry1ab*, *cry1ba*, and *cry1bb*, likely derived from TGD. These results are consistent with our previous study [36]. In addition, spotted gar also contain one *cry2* gene, the ortholog of mammalian *Cry2*; and one *cry3* gene, which mammals do not have.

In the *par* gene family, spotted gar has one *tef* and one *hlf* gene, like humans (both genes with TGD ohnologs). The potential ortholog of *dbp* is present as an unannotated sequences on an unassembled scaffold (JH591448.1: 146,984 to 147,190) in the current version (LepOcu1) of the spotted gar genome (reciprocal best blast hit with coelacanth *dbp* and nearest four neighbors on one side being orthologs of the four nearest neighbors of zebrafish *dpba*).

### 3.2. Evolution of nfil3 Genes

The phylogenetic analysis of *nfil3* family genes showed that there is only one *nfil3* ortholog in flies (Figure 2A), and this gene family can be classified into three main subclades in vertebrates. The *nfil3-1* gene is shared among most chordates, with the exception of its loss in a few teleost fish (fugu and medaka). Almost all fish, including coelacanth and spotted gar, have the other two *nfil3* members, *nfil3-2* and *nfil3-3*. Intriguingly, lizards have only *nfil3-2*, while frogs have only *nfil3-3*. However, both *nfil3-2* and *nfil3-3* are completely lost in mammals and birds. Further, *nfil3-2* and *nfil3-3* were duplicated to *nfil3-2a/nfil3-2b* and *nfil3-3a/nfil3-3b* in teleost fish via TGD, respectively. As expected, some of these *nfil3* duplicates were lost in some fish lineages—for instance, *nfil3-2a* and *nfil3-3a* in fugu, *nfil3-3a* in tetraodon, *nfil3-3b* in medaka, and *nfil3-3b* in cave fish were all gone. Moreover, zebrafish and cave fish possess additional *nfil3-1b.1* and *nfil3-1b.2* genes, while tilapia and platyfish only have *nfil3-1b.1*, and tetraodon only has *nfil3-1b.2*, which appears to derive from local tandem duplication and cannot be found in other species (Figure 2C). Finally, phylogenetic analysis also showed that *Nfil3-1* was duplicated to *Nfil3-1a* and *Nfil3-1b* in chickens and ducks.

Two or more orthologous genes linked in a single chromosome or a chromosomal fragment in each of two or more different species define a conserved syntenic region [37,38]. The conserved syntenic analyses provide important evidence for the duplication of genes and genomes. To further study the evolutionary relationships of the *nfil3* family members, we also conducted a syntenic analysis. Using the Genomicus Database, we determined the orthologs of *nfil3* in vertebrates. *nfil3-1* is conserved from spotted gar and coelacanth to humans (Figure 2B). We also observed highly conserved synteny in fugu and medaka genomes, although the two teleost fish do not possess the *nfil3-1* gene. In addition, some teleosts contain *nfil3-1b.1* and *nfil3-1b.2* members, which are linked in the same chromosome. The conserved synteny structures are found in all teleost fish, although some species have lost their duplicates (Figure 2C). Moreover, *nfil3-2* is linked to *nfil3-3* in the same chromosome. A similarly conserved synteny is also observed in species from coelacanth to reptiles, although there is only one of *nfil3-2* and *nfil3-3* maintained in amphibians or reptiles (Figure 2D). Interestingly, even though mammals do not possess any *nfil3-2* or *nfil3-3* genes, we still observed highly conserved synteny in their genomes. On the other hand, the exon structures of the three *nfil3* subclade genes are also similar, and most of the *nfil3* genes have only one exon.

### 3.3. Evolution of Cry Genes

The *cry1a* and *cry1b* genes of spotted gar are clustered with human *CRY1* genes (Figure 3A). Phylogenetic analysis showed that all teleost fish *cry1* genes can be classified into two subclades, *cry1a* and *cry1b*. Further, *cry1a* and *cry1b* have been duplicated in most teleost fish, giving rise to *cry1aa*, *cry1ab*, *zcry1ba*, and *cry1bb*, as reported previously [36]. In addition to tetraodon, which has a *cry2a*/*cry2b* duplicate, all other fish each have only one *cry2* gene, forming a monophyletic group with human *CRY2* genes (Figure 3A). In a separate monophyletic clade, spotted gar, teleost fish, turtles, and birds each have one *cry3*, which cannot be found in mammals (Figure 3A). 

We also observed conserved syntenic structures surrounding *cry1a* and *cry1b* genes in vertebrate genomes (Figure 3B). The exon–intron structures of *cry1, cry2*, and *cry3* are highly conserved, with the same coding exon numbers and function domain location found in spotted gar, teleost fish, birds, and humans (Figure 3C). 

### 3.4. Evolution of the Par Family Genes

The *par* gene family contains three members: *tef*, *hlf*, and *dbp*. While spotted gar has one *tef* and one *hlf* orthologs as mammals, most of the teleost fish have two *tef* genes (*tefa*, *tefb*) and two *hlf* genes (*hlfa*, *hlfb*), which form monophyletic groups with spotted gar orthologs, respectively (Figure 4A). The potential gar ortholog of *dbp* is present as a partial unannotated sequence (Scaffold JH591448.1:146984-147190, 69aa) in the spotted gar genome (reciprocal best blast hit with coelacanth *dbp* and the nearest four neighbors on one side being orthologs of the four nearest neighbors of zebrafish *dpba*). *Dbp* orthologs are also conserved in most vertebrates. There is only one *Dbp* gene in mammals, which was duplicated to *dbpa* and *dbpb* in teleost fish.

The exon structures of *hlf* genes are conserved from spotted gar to humans (four coding exons), but the structure of *tef* in gar (six coding exons) is different between zebrafish and humans (four coding exons) (Figure 4B).

### 3.5. Preservation of Regulatory Elements in Duplicated Circadian Clock Genes

Although the genetic regulation of circadian rhythms appears to be similar between fish and mammals, the numbers of gene copies involved in this process vary among these groups. To further study the regulation of the duplicated genes, we compared the transcription factor binding elements of *per* genes between spotted gar, teleost fish, and mammals. There are some motifs, such as NFE2L1::MafG, Prrx2 and SOX10, in upstream regions of the orthologus genes from spotted gar to mammals. Interestingly, we found that the Ahr::Arnt motif exists in unduplicated *per1* and *per2* ortholog genes in spotted gar, some teleosts, and mammals, but this motif only exists in one of the *per1* or *per2* duplicated pairs in teleost fish (Figure 5). For instance, while *per1a* in zebrafish (Figure 5A) and *per2a* in fugu, medaka, and tetraodon (Figure 5B) have Ahr::Arnt motifs, the *per1b* (Figure 5A) and *per2b* (Figure 5B) genes in these species do not, thereby implicating possible cases predicted by the duplication-degeneration-complementation (DDC) model. The results of all regulatory elements in *per1* and *per2* genes are shown in Appendix A.

## 4. Discussion

Teleost fish provide an attractive model for studying a multitude of questions related to evolution. This may be linked to the apparent, considerable, plasticity of their genome, manifested, for example, by a high variability in genome size and chromosome number [39]. Particularly, there is now substantial evidence that an ancient event of genome duplication provided the evolutionary framework for the diversification of gene functions and for speciation in fish [22,40]. Phylogenetic analyses have suggested that spotted gar occupy a clade of ancient ray-finned fish that diverged from the teleost lineage after the divergence of the bichir (*Polypterus sp.*) lineage [25,26,41,42,43]. Among species occupying this pre-TGD clade, spotted gar appear to be the most suitable for studies of development, genomics, and physiology [16,24,25]. Circadian rhythms are endogenous rhythms that are observed in a wide range of life forms, and circadian oscillators exhibit remarkable conservation of function across wide evolutionary time spans [44]. Our analyses of spotted gar circadian clock genes provide insight into the evolution and evolutionary impact of circadian clock genes in fish genomes and in vertebrates. 

In this study, we studied the evolutional relationship of *nfil3* genes in vertebrates. The phylogenetic and the syntenic analyses support the notion that the teleost fish and tetrapod *nfil3* have a common ancestor. There is only one *nfil3* ortholog in lampreys, and *nfil3* genes have increased to three in coelacanth and spotted gar. During vertebrate evolution, the second round of vertebrate genome duplication (VGD) arose after the divergence between agnatha and teleost fish. The most parsimonious explanation is that the ancestral *nfil3* gave rise to *nfil3-1/2* and *nfil3-3/4* by tandem duplication. Then, the clusters were duplicated to *nfil3-1, nfil3-2, nfil3-3*, and *nfil3-4* genes in the VGD2, with the quick loss of some orthologs (Figure 6). The *nfil3-1* is conserved in all vertebrates, with the exception of their loss in a few teleost fish. Although fugu and medaka do not possess *nfil3-1*, we still observed highly conserved synteny surrounding *nfil3-1* in their genomes, implying that gene loss occurred in the two species. Further, the *nfil3-1b.1/nfil3-1b.2* linkage was only found in teleost fish, and the conserved synteny surrounding *nfil3-1b.1/nfil3-1b.2* was observed in all teleost fish. It seems that *nfil3-1b.1* and *nfil3-1b.2* were derived from local tandem duplication before teleost TGD. On the other hand, *nfil3-2* is linked to *nfil3-3* in the same chromosome in teleost fish and tetrapod lineage, except for mammals and birds. Amphibians retained only *nfil3-3*, while reptiles retained only *nfil3-2*. It is possible that one of the linked chromosome fragments has been lost in amphibians and reptiles, respectively. The highly conserved synteny surrounding *nfil3-2* and *nfil3-3* was also observed in mammalian genomes. Thus, it is likely that both *nfil3-2 and nfil3-3* existed in tetrapod ancestors and were lost in mammals during evolution. Finally, the *nfil3-2a/2b* and *nfil3-3a/3b* in teleost fish were likely derived via the TGD (Figure 6).

The *Cry* genes had already evolved before the origin of eukaryotic organisms. The phylogenetic and syntenic analyses also suggested that the teleost fish and tetrapod *Cry* have a common ancestor. The vertebrate *Cry1* group and *Cry2* group form a sister clade, suggesting that they were derived from the duplication of a common ancestral gene. It seems that, during first round of VGD, the ancestral *Cry* gave rise to *Cry12* and *Cry34*. During the second round of VGD, *Cry12* gave rise to *Cry1* and *Cry2*, while *Cry34* gave rise *Cry3* and *Cry4*, but *Cry4* was quickly lost. *Cry1* and *Cry2* have been preserved in all vertebrates, while *Cry3* genes are preserved only in the teleost and tetrapod lineages but not in mammals. In the teleost fish, *cry1* generated *cry1a* and *cry1b* by local gene duplication before teleost genome duplication (TGD). The TGD generated extra copies of *cry1a* in teleost fish. For instance, zebrafish has retained *cry1aa*, *cry1ab*, *cry1ba*, *cry1bb*, *cry2*, and *cry3* genes, which are consistent with our previous findings [36].

It is important to understand the evolutionary fates of circadian clock genes. According to the duplication-degeneration-complementation (DDC) model, one of the potential fates of duplicate gene pairs with multiple regulatory regions is that each duplicate may experience a loss or reduction of its expression for different subfunctions by degenerative mutations [22]. The combined action of both polygene copies is necessary to fulfill the requirements of the ancestral locus (subfunctionalization). If this happens, then complementation of the subfunctions between duplicate genes will preserve both partially degenerated copies [22]. The Ahr::Arnt motif DNA binds with protein interactions of the AHR/ARNT heterodimer [45]. ARNTL (BMAL) could interact with the PER protein. It seems that *per1a/per1b* or *per2a/per2b* have preserved some functions compared to their ancestors (subfunctionalization). Interesting, the duplicated copies of the *per1* gene found in zebrafish showed distinct patterns of temporal and spatial expression [46]. The *per1a* gene is expressed in the retina and in both the telencephalon and the diencephalon of the forebrain in a dark environment, when there is no detectable expression of *per1b*. However, the expression of *per1a* is significantly up-regulated, and the *per1b* gene is constitutively expressed throughout the head region when the fish are in a light environment [46]. This pattern represents a clear example of subfunctionalization, in which each daughter copy adopts part of the function of the parental gene [27]. Our analysis of the regulatory elements in *per* genes may provide evidence that the mutations of these binding motifs caused gene subfunctionalization.

Compared to mammals, the extra copies of core clock genes in spotted gar were found primarily in the *nfil3* and *cry* families. It appears that these genes duplicated before the TGD. In mammals, *NFIL3* is a homologue of *vrille* (*vri*), which functions as a key negative component of the circadian clocks in *Drosophila* [47,48,49,50,51]. In mammals, NFIL3 was shown to interact with PER2 as well as CRY2, suggesting that NFIL3 binds to PER2 or CRY2 and contributes to mammalian circadian regulation [52,53]. Although there is a remarkable conservation of function in the circadian molecular machinery in vertebrates, studies have suggested differences in circadian regulation between fish and mammals [54,55,56]. Teleost fish exhibit an amazing level of biodiversity. In contrast to mammalian genomes, teleost genomes also contain multiple gene families. The ancient event of genome duplication could provide the evolutionary framework for the diversification of gene functions and for speciation in fish [14]. The duplication of a gene can relieve selective pressures from the duplicates because one copy can compensate for deleterious mutations in the other. The existence of paralogs enables individual members of the gene family to specialize and extend their original ancestral roles, thereby generating the potential for the evolution of complex regulatory mechanisms [44]. Hence, it is tempting to speculate that the duplication of core clock genes (such as *nfil3* and *cry*) in teleost fish diversified the circadian regulatory mechanisms in fish.

In conclusion, we uncovered 26 circadian clock genes from 11 families among spotted gar, and nine of these 11 families of spotted gar circadian clock genes possessed the same number as humans, without an extra duplicate copy. The spotted gar contains *nfil3-1, nfil3-2.1*, and *nfil3-2.2*. While *nfil3-1* is the ortholog of mammalian *Nfil3-1*, the *nfil3-2* and *nfil3-3* orthologs have been lost in mammals. For the *cry* gene family, spotted gar has two *cry1* genes, *cry1a*, and *cry1b*, which likely derived from local (tandem) duplication in their ancestor and are co-orthologs of mammalian *Cry1*. Similar to teleosts, spotted gar also contain *cry3*, which mammals do not have. In addition, we also found that the transcription factor binding element (Ahr::Arnt) is retained only in one of the *per1* or *per2* paralogs in the teleost fish, suggesting that subfunctionalization occurred in the teleost fish *per* family. These results strongly support the notion that spotted gar genomically resembles humans to a great extent (without the TGD), but still contain extra copies of genes that mammals lack.

## Figures and Tables

**Figure 1 genes-10-00622-f001:**
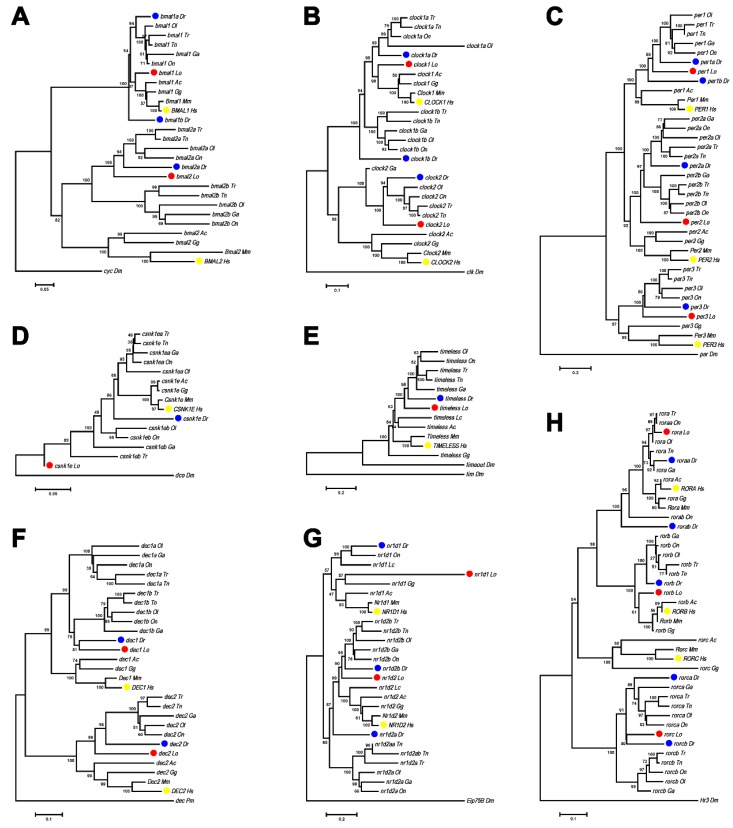
The phylogenetic trees of circadian clock genes. (**A**) *bmal* genes; (**B**) *clock* genes; (**C**) *per* genes; (**D**) *csnk1e* genes; (**E**) *tim* genes; (**F**) *dec* genes; (**G**) *nr1d* genes; (**H**) *ror* genes. The trees were constructed by the neighbor-joining method with MEGA6 [33], and the numbers on the nodes are percent bootstrap values based on 1000 pseudoreplications. The outgroups are ortholog genes in fly or lamprey. Red dots indicate spotted gar genes, blue dots indicate zebrafish genes, and yellow dots indicate human genes. The sequences used are listed in Appendix A.

**Figure 2 genes-10-00622-f002:**
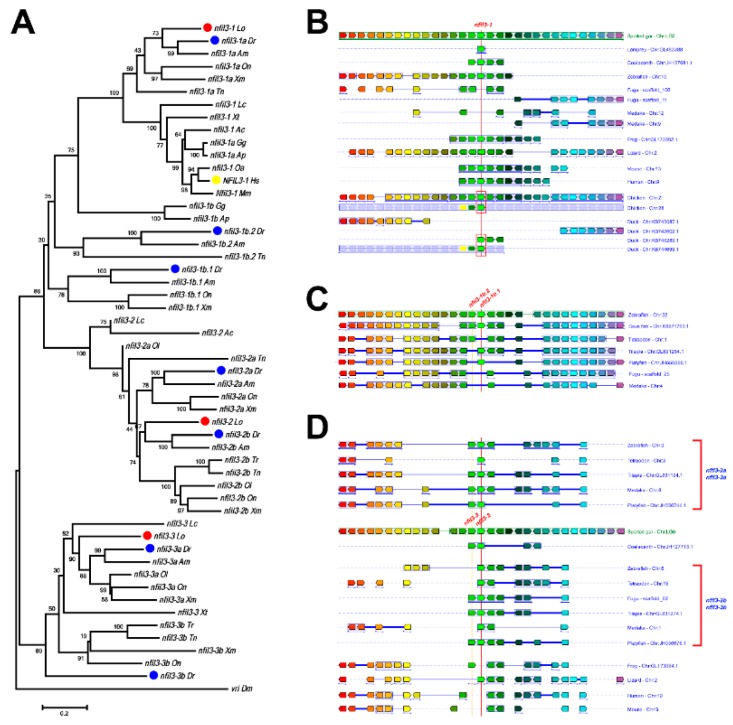
The phylogenetic and conserved syntenic analyses of the *nfil3* family. (**A**) Phylogenetic trees based on sequences of *nfil3* genes. The tree was constructed by the neighbor-joining method with MEGA6 [33], and the numbers on the nodes are the percent bootstrap values based on 1000 pseudoreplications. The outgroup is fly *vri*. Red dots indicate spotted gar genes, blue dots indicate zebrafish genes, and yellow dots indicate human genes. (**B**) The conserved synteny surrounding *nfil3-1* genes in the chromosomes of lampreys, coelacanths, spotted gar, zebrafish, medaka, fugu, frogs, lizards, chickens, ducks, mice and humans was displayed with Genomicus [35]. (**C**) The conserved synteny of *nfil3-1b.1* and *nfil3-1b.2* genes in the chromosomes of zebrafish, cave fish, tilapia, platyfish, medaka, tilapia, and fugu. (**D**) Conserved synteny of *nfil3-2* and *nfil3-3* genes in chromosomes of coelacanth, spotted gar, zebrafish, medaka, teraodon, fugu, tilapia, platyfish, frogs, lizards, mice, and humans. The chromosome number or linkage group number is shown to the right of the chromosomes. The same colored rectangles represent the orthologous genes. Red and blue boxes indicate the duplicated genes.

**Figure 3 genes-10-00622-f003:**
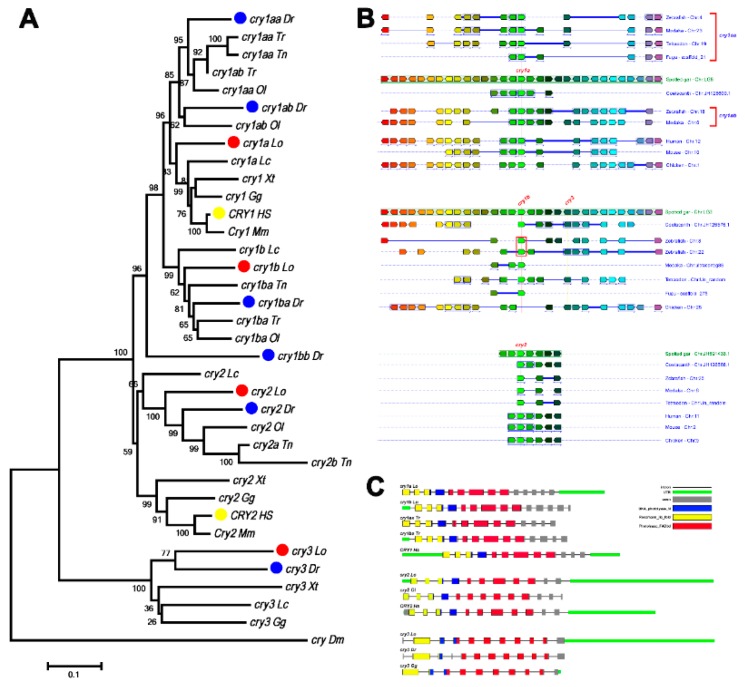
The phylogenetic, conserved syntenic, and exon structural analyses of the *cry* family. (**A**) Phylogenetic trees based on sequences of *cry* genes. The tree was constructed by the neighbor-joining method with MEGA6, and the numbers on the nodes are the percent bootstrap values based on 1000 pseudoreplications. The outgroup is *Drosophila melanogaster cry*. Red dots indicate spotted gar genes, blue dots indicate zebrafish genes, and yellow dots indicate human genes. (**B**) The conserved synteny surrounding the *cry1a*, *cry1b/cry3*, and *cry2* genes in chromosomes of coelacanth, spotted gar, zebrafish, medaka, teraodon, fugu, chickens, mice, and humans. The chromosome number or linkage group number is shown to the right of the chromosomes. The same colored rectangles represent the orthologous genes. Red and blue boxes indicate the duplicated genes. (**C**) The exonic structures of *cry* genes in spotted gar, medaka, fugu, chickens and humans. The boxes represent the exons. The size of each exon is drawn to scale.

**Figure 4 genes-10-00622-f004:**
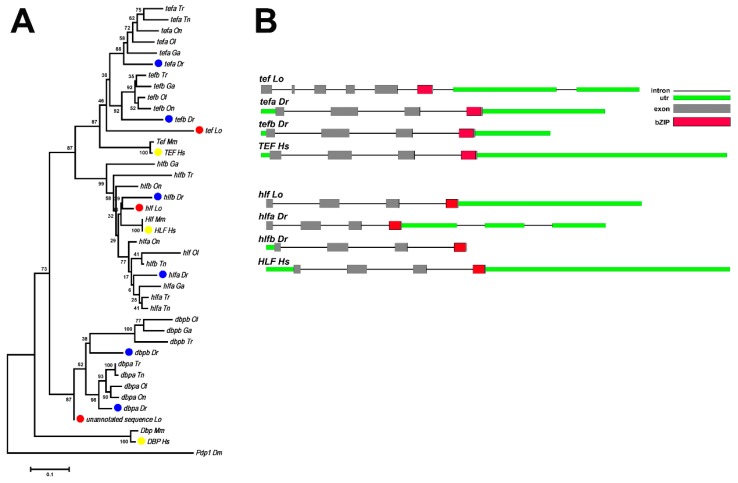
The phylogenetic and exon structural analyses of the *par* family. (**A**) Phylogenetic trees based on sequences of *PAR* family genes. The tree was constructed by the neighbor-joining method with MEGA6, and the numbers on the nodes are the percent bootstrap values based on 1000 pseudoreplications. The outgroup is *Drosophila melanogaster pdp1*. Red dots indicate spotted gar genes, blue dots indicate zebrafish genes, and yellow dots indicate human genes. (**B**) The exonic structures of *par* family genes in spotted gar, zebrafish, and humans. The boxes represent the exons. The size of each exon is drawn to scale.

**Figure 5 genes-10-00622-f005:**
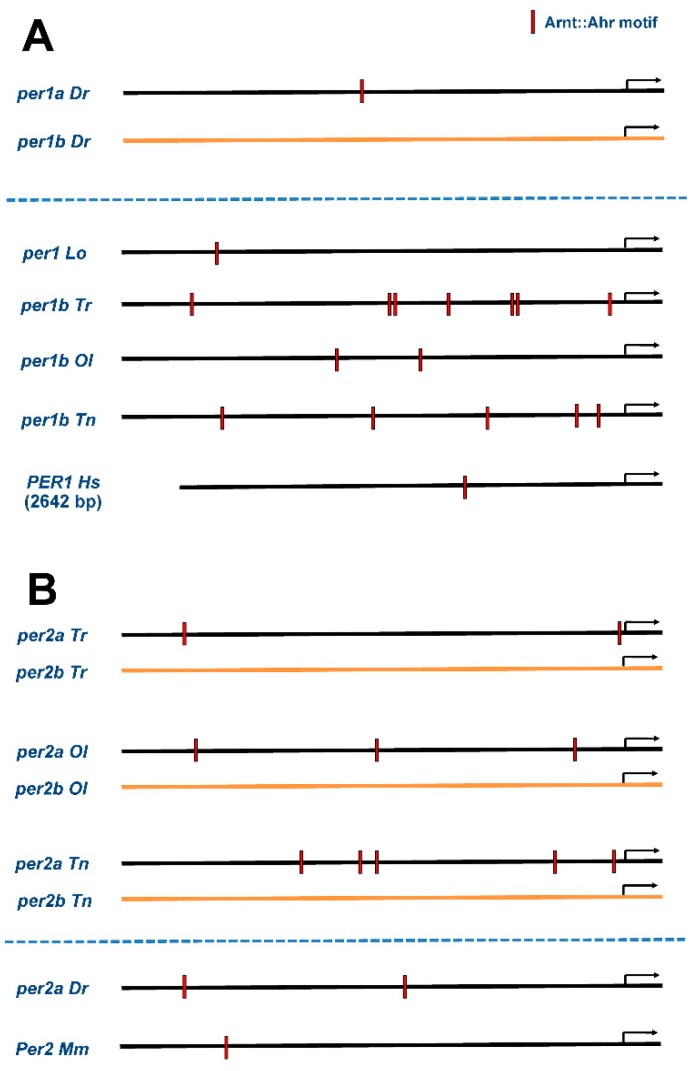
The transcription factor binding element (Ahr::Arnt) in upstream regions of duplicated *Per* genes from spotted gar to mammals. The complementary presence of the Ahr::Arnt motif in duplicated *Per1* (**A**) or *Per2* (**B**) paralog genes provides possible cases predicted by the duplication-degeneration-complementation (DDC) model. The upstream region of each gene is 3000 bp in length, with the exception of that of human *PER1*, which is 2642 bp only. The transcription factor binding elements were predicted using the JASPAR dataset, with a profile score threshold of 99%. The bisque line represents duplication in the lineage, and the dotted line separates the duplicated genes and unduplicated genes.

**Figure 6 genes-10-00622-f006:**
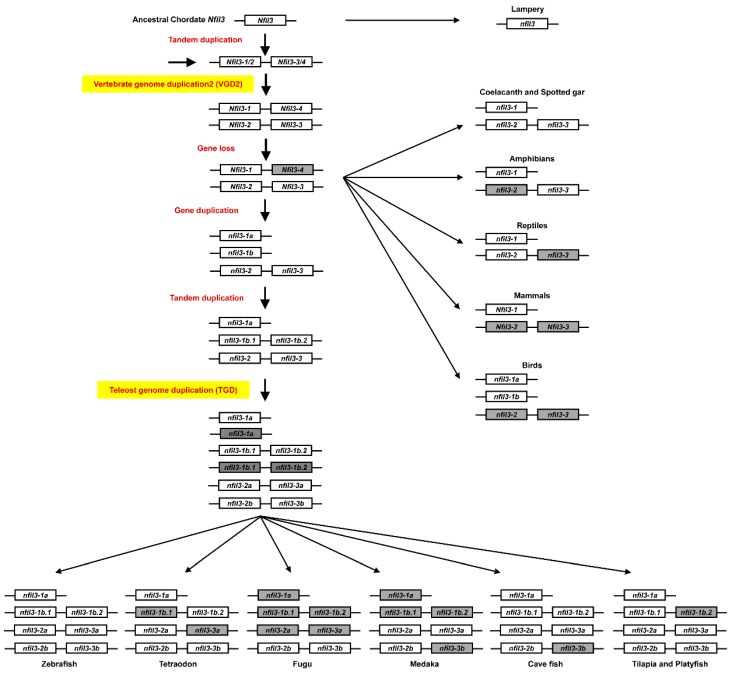
A hypothetical model for the evolution of *nfil3* genes in vertebrates. The teleost fish and tetrapod *nfil3* had a common ancestor. The ancestral *nfil3* gave rise to *nfil3-1/2* and *nfil3-3/4* by tandem duplication. During the second round of vertebrate genome duplication, the cluster was duplicated to *nfil3-1, nfil3-2, nfil3-3*, and *nfil3-4* genes, with the quick loss of *nfil3-4*. The *nfil3-1* is conserved in all vertebrates, with the exception of losses in a few teleost fish. The *nfil3-1b.1* and *nfil3-1b.2* linkage has been preserved only in teleost fish. The *nfil3-2* and *nfil3-3* linkage was conserved in the teleost fish and tetrapod lineages. The teleost-specific genome duplication produced the *nfil3-2a/nfil3-2b* duplicate, which was observed in zebrafish, medaka, tetrodon, tilapia, platyfish, and cave fish; and the *nfil3-3a/nfil3-3b* duplicate, which was observed in zebrafish, tilapia, and platyfish. However, amphibians retained only *nfil3-3* and reptiles retained only *nfil3-2*. Then, *Nfil3-2* and *Nfil3-3* were completely lost in the lineages of mammals and birds during evolution. The grey box means gene loss.

**Table 1 genes-10-00622-t001:** Circadian clock genes in spotted gar [16].

Gene Names	Ensembl Gene ID	Protein Length (aa)	Genome Location
*bmal1*	ENSLOCG00000003999	678	Chromosome LG27: 8,317,763–8,344,549
*bmal2*	ENSLOCG00000015224	639	Chromosome LG8: 3,189,570–3,226,867
*clock1*	ENSLOCG00000014043	744	Chromosome LG4: 72,323,329–72,339,605
*clock2*	ENSLOCG00000014750	886	Chromosome LG7: 42,295,248–42,323,335
*cry1a*	ENSLOCG00000015272	647	Chromosome LG8: 4,053,475–4,074,670
*cry1b*	ENSLOCG00000011417	675	Chromosome LG3: 32,901,867–32,938,020
*cry2*	ENSLOCG00000014655	569	Scaffold JH591436.1: 96,765–111,323
*cry3*	ENSLOCG00000011465	586	Chromosome LG3: 33,014,383–33,053,389
*per1*	ENSLOCG00000013344	1445	Chromosome LG2: 58,185,728–58,201,428
*per2*	ENSLOCG00000004441	1385	Chromosome LG14: 7,862,125–7,881,568
*per3*	ENSLOCG00000002607	1165	Chromosome LG25: 4,785,705–4,800,981
*csnk1e*	ENSLOCG00000011701	273	Chromosome LG12: 35,099,178-35,103,163
*dec1*	ENSLOCG00000010962	409	Chromosome LG5: 27,670,723–27,673,540
*dec2*	ENSLOCG00000015327	422	Chromosome LG8: 4,744,466–4,746,682
*nfil3-1*	ENSLOCG00000008217	443	Chromosome LG2: 25,281,929–25,283,356
*nfil3-2*	ENSLOCG00000018299	544	Chromosome LG6: 17,036,778–17,038,412
*nfil3-3*	ENSLOCG00000018298	394	Chromosome LG6: 17,009,772–17,010,956
*nr1d1*	ENSLOCG00000006223	362	Chromosome LG4: 15,608,213–15,662,480
*nr1d2*	ENSLOCG00000006818	604	Chromosome LG11: 20,445,844–20,461,290
*tef*	ENSLOCG00000011595	323	Chromosome LG12: 34,848,929–34,860,886
*hlf*	ENSLOCG00000012233	298	Chromosome LG10: 33,240,269–33,260,238
*dbp*		69	Scaffold JH591448.1:146984–147190
*rora*	ENSLOCG00000014779	519	Chromosome LG3: 53,154,612–53,409,505
*rorb*	ENSLOCG00000009712	462	Chromosome LG2: 34,273,732–34,319,408
*rorc*	ENSLOCG00000006502	467	Chromosome LG19: 9,503,786–9,519,701
*timeless*	ENSLOCG00000004180	1225	Chromosome LG4: 11,892,308–11,918,271

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
