# Peer review of "The Molecular Evolution of Circadian Clock Genes in Spotted Gar (*Lepisosteus oculatus*)"

_genes, 2019, doi:10.3390/genes10080622_

Round 1

Reviewer 1 Report

This is an interesting study to investigate the circadian clock genes, and compare the difference of these genes between spotted gar vs. mammals. It is well written with interesting results. It is acceptable.  

Author Response

We appreciate reviewer very much for the positive comments.

Reviewer 2 Report

I think this study sheds some light to very important problem. But there is a point that, I suppose, should be improved.

I do not know very much about clock genes, so it's unclear for me why have you chosen exactly these gene families. It would be better if you explain it and describe their functions in detail in "Introduction" and also pay more attention to them in "Discussion".

I also have some questions about the results representation.

Figures 2 (B-D), 3 (B-D) and S1-5: what does the blue lines (thin, bold and dotted) represent?

There are no revised names for timeless gene family in Table S1. If no name in this family have been changed, maybe it would be better to mention it in "Materials and methods".

Author Response

Thanks very much for the reviewer' comments concerning our manuscript.

Point 1: I do not know very much about clock genes, so it's unclear for me why have you chosen exactly these gene families. It would be better if you explain it and describe their functions in detail in "Introduction" and also pay more attention to them in "Discussion".
Response 1: The 11 families have contained the main circadian clock genes members. According to the reviewer's suggestion, we have re-written the Introduction and Discussion part to introduce these genes in more detail in the revision.

Point 2: Figures 2 (B-D), 3 (B-D) and S1-5: what does the blue lines (thin, bold and dotted) represent?
Response 2: It is really true as reviewer suggested that these labels should been described in the manuscript. We have added a detailed description of these labels in "Materials and Methods (2.3. Conserved synteny analysis)".

Point 3: There are no revised names for timeless gene family in Table S1. If no name in this family have been changed, maybe it would be better to mention it in "Materials and methods".
Response 3: There is only one tim ortholog gene in each species of vertebrate. So we don't change the names of timeless genes. We have added the description in "Materials and Methods (2.1. Data sets and Phylogenetic Analysis)".

Special thanks to you for your good comments!

This manuscript is a resubmission of an earlier submission. The following is a list of the peer review reports and author responses from that submission.